# Modeling Seasonal Fire Probability in Thailand: A Machine Learning Approach Using Multiyear Remote Sensing Data

Enikoe Bihari [1], Karen Dyson [1,*], Kayla Johnston [1], Daniel Marc G. dela Torre [1], Akkarapon Chaiyana [1], Karis Tenneson [1], Wasana Sittirin [1], Ate Poortinga [1], Veerachai Tanpipat [1,2], Kobsak Wanthongchai [3], Thannarot Kunlamai [1], Elijah Dalton [1], Chanarun Saisaward [1], Marina Tornorsam [4], David Ganz [1,4] and David Saah [1,5]

[1] Spatial Informatics Group, Pleasanton, CA 94566, USA; ebihari@sig-gis.com (E.B.); kjohnston@sig-gis.com (K.J.); dtorre@sig-gis.com (D.M.G.d.T.); ktenneson@sig-gis.com (K.T.); wsittirin@yahoo.com (W.S.); fforvrc@ku.ac.th (V.T.); tkunlamai@sig-gis.com (T.K.); edalton@sig-gis.com (E.D.); csaisaward@sig-gis.com (C.S.); david.ganz@recoftc.org (D.G.); dsaah@sig-gis.com (D.S.)

[2] The Upper ASEAN Wildland Fire Special Research Unit, Forestry Research Center, Faculty of Forestry, Kasetsart University, Bangkok 10900, Thailand

[3] Department of Silviculture, Faculty of Forestry, Kasetsart University, Bangkok 10900, Thailand; fforksw@ku.ac.th

[4] Regional Community Forestry Training Center for Asia and the Pacific (RECOFTC), Bangkok 10900, Thailand; marina.tornorsam@recoftc.org

[5] Department of Environmental Science, University of San Francisco, San Francisco, CA 94117, USA

\* Correspondence: kdyson@sig-gis.com

## Highlights

### What are the main findings?

- Anthropogenic fire patterns in northern Thailand are captured well by pairing proven Random Forest modelling methods with a localized model development process, including temporal data disaggregation, representative reference data sampling, and empirical predictor variable selection.
- This method represents a scalable advancement in wildfire probability mapping using open-source tools for data-constrained landscapes.

### What is the implication of the main finding?

- Modelling fire probability seasonally using annually paired fire occurrences and predictor variables allows researchers and managers to explore year-to-year variability in fire patterns, which is particularly critical for pre-fire-season resource allocation.
- Annual updates to fire probability maps using this method fill a critical resource gap for fire prevention and response in northern Thailand, helping fire managers to optimize fire management at every stage of planning.

## Abstract

Seasonal fires in northern Thailand are a persistent environmental and public health concern, yet existing fire probability mapping approaches in Thailand rely heavily on subjective multi-criteria analysis (MCA) methods and temporally static data aggregation methods. To address these limitations, we present a flexible, replicable, and operationally viable seasonal fire probability mapping methodology using a Random Forest (RF) machine learning model in the Google Earth Engine (GEE) platform. We trained the model on historical fire occurrence and fire predictor layers from 2016–2023 and applied it to 2024 conditions to generate a probabilistic fire prediction. Our novel approach improves upon existing operational methods and scientific literature in several ways. It uses a more representative

sample design which is agnostic to the burn history of fire presences and absences, pairs fire and fire predictor data from each year to account for interannual variation in conditions, empirically refines the most influential fire predictors from a comprehensive set of predictors, and provides a reproducible and accessible framework using GEE. Predictor variables include both socioeconomic and environmental drivers of fire, such as topography, fuels, potential fire behavior, forest type, vegetation characteristics, climate, water availability, crop type, recent burn history, and human influence and accessibility. The model achieves an Area Under the Curve (AUC) of 0.841 when applied to 2016–2023 data and 0.848 when applied to 2024 data, indicating strong discriminatory power despite the additional spatial and temporal variability introduced by our sample design. The highest fire probabilities emerge in forested and agricultural areas at mid elevations and near human settlements and roads, which aligns well with the known anthropogenic drivers of fire in Thailand. Distinct areas of model uncertainty are also apparent in cropland and forests which are only burned intermittently, highlighting the importance of accounting for localized burning cycles. Variable importance analysis using the Gini Impurity Index identifies both natural and anthropogenic predictors as key and nearly equally important predictors of fire, including certain forest and crop types, vegetation characteristics, topography, climate, human influence and accessibility, water availability, and recent burn history. Our findings demonstrate the heavy influence of data preprocessing and model design choices on model results. The model outputs are provided as interpretable probability maps and the methods can be adapted to future years or augmented with local datasets. Our methodology presents a scalable advancement in wildfire probability mapping with machine learning and open-source tools, particularly for data-constrained landscapes. It will support Thailand's fire managers in proactive fire response and planning and also inform broader regional fire risk assessment efforts.

**Keywords:** seasonal fire probability; annual fire prediction; Northern Thailand; machine learning; random forest; fire risk modeling; historical fire data; remote sensing; Google Earth Engine; burn scar

## 1. Introduction

Thailand experiences seasonal forest and agricultural fires beginning in January or February and continuing until the onset of the rainy season in May [1–5]. Predicting seasonal wildfires in this time period is critical to mitigating the negative environmental and human health impacts, including landscape degradation and high levels of hazardous PM 2.5 air pollution [1,6–11].

Wildfires in Thailand are challenging to predict as they are largely driven by anthropogenic, rather than natural factors, including cultural, economic, and legal activities [12–14]. These anthropogenic influences not only drive current fire events but also initiate feedback mechanisms that create conditions conducive to future wildfires (Supplement Figure S1) [15–19]. More specifically, wildfires most commonly result from escaped intentional burns, carelessness, or arson, though some are ignited by lightning [1,18]. Intentional burns are used for clearing vegetation for agriculture and hunting, clearing understory for non-timber forest product (NTFP) collection, stimulating tree regeneration, eliminating crop residue from rice, maize, and sugarcane plantations, and regenerating grass on livestock grazing land [16,17,19–25]. Only 5% of forest fires between 2001–2021 in Thailand were associated with complete forest loss, suggesting that forests are degraded but not completely cleared by most types of fire events [2].

Fire feedback mechanisms are both natural and anthropogenic. The severity of wildfires is seasonally influenced by El Niño and La Niña (ENSO) [25] and directionally exacerbated by population growth and the growing need for land for cultivation [23,26,27]. During the 10 years between 2014–2023, over 67% of all burned area burned at least twice and over 22% burned at least 5 times (on average every other year), and in some rare cases repeated burns even occurred within the same year [26]. This indicates that land burned once is more likely to burn again, possibly multiple times, within the next decade. When land is initially cleared with fire for agricultural use, it often enters a multi-year cultivation cycle in which crop residue will be repeatedly burned, simultaneously increasing accessibility and economic incentives to convert neighboring land to agriculture as well [15]. Certain forest types, such as dry dipterocarp, bamboo, and teak forests, are naturally more fire-prone and more frequently burned for NTFP collection or tree regeneration [14,15,17,18]. Many of these ecosystems are maintained naturally by edaphic, climatic, and ecological factors [27–30] while others are likely the result of human-facilitated encroachment on other ecosystems [27,28,31,32] and may eventually return to wet forest types if left unburned [28,33]. In such cases, when wet tropical forests are degraded to drier forest types through burning, they are more likely to be burned again for NTFP production; meanwhile, they accumulate increasingly flammable fuels, necessitating continued burning for fuel management that may not have been necessary before [15].

Together, these complexities make it difficult to accurately predict and manage fires in Thailand. Mapping wildfire probability and risk are critical components of wildfire management. Fire risk is the combination of fire probability, intensity, exposure, and susceptibility [34]. Fire risk maps predict where fires will occur and how dangerous they will be [34–36]. In this study, we will refer to "fire probability" to mean the likelihood of a fire occurring and "fire risk" to mean the comprehensive quantification of probability, intensity, exposure, and vulnerability. Note that in the Thai language scientific and government literature frequently uses "fire risk" to refer to what English speaking fire managers define as "fire probability". This study addresses fire probability, which is foundational to quantifying fire risk but does not provide a complete assessment of fire risk.

Existing work on fire probability mapping in Thailand and peninsular Southeast Asia includes operational methods and scientific studies, which are described in detail in Supplement Tables S1 and S2. Operational methods in Thailand generally use multi-criteria analysis (MCA) to evaluate fire probability, which is a method that weights predictor variables based on expert opinion or statistical distributions and combines them into a final score (Supplement Table S1) [1,10,37]. Current operational map products include a triennial map published by the Department of National Park, Wildlife and Plant Conservation (DNP) for internal use and a weekly map published by the Geo-Informatics and Space Technology Development Agency (GISTDA) for public use [1,10,37]. These are generated using MCA, and predictor variables include burn history and frequency, land use and land cover (LULC), vegetation and water indices, fire season weather, infrastructure and accessibility, and topography [1,10,37]. Additionally, both the DNP and the Royal Forest Department (RFD) use different iterations of the Thai Fire Danger Rating System (FDRS), generating daily Fine Fuel Moisture Code (FFMC) and Fire Weather Index (FWI) maps which indicate the daily danger of fire occurrence (thematically related but not fully analogous to fire probability maps) [25,38,39]. These maps are produced using daily weather forecasts and mathematical relationships between fire occurrence and meteorological parameters [25]. The Thai FDRS was adapted from the Canadian FDRS, and still needs significant calibration to the fuel and weather conditions of Thailand [15].

Scientific literature exploring fire probability mapping in peninsular Southeast Asia uses either MCA [40–44], probabilistic statistics (PS) [45,46], or machine learning

(ML) [47–52], approaches, with MCA and ML being the most common (Supplement Table S2). The most common MCA approach is Analytical Hierarchy Process (AHP), while many different ML approaches are used, including Random Forest (RF), Bayes Network (BN), Naïve Bayes (NB), and Support Vector Machine (SVM). These studies have generally been developed at either coarse scales for all of Southeast Asia or at fine scales for small pilot regions such as individual national parks or provinces in Thailand, Vietnam, and Malaysia [40,41,43–52]. Predictor variables commonly include burn history and frequency, LULC, vegetation and water indices, fire season weather, infrastructure and accessibility, and topography [40–52]. Thaewthatum et al., 2017 used MCA to map fire probability for a study area in Northern Thailand nearly identical to ours [42], while Nuthammachot and Stratoulias 2019, Nuthammachot and Stratoulias 2021, and Burapapol and Nagasawa et al., 2017 used MCA for much smaller regions within Thailand [40,41,43]. He et al., 2021 compared multiple ML methods for a study area that spanned Southeast Asia (including Thailand), and Ahmad et al., 2019 used PS for an even larger study area that included both South and Southeast Asia (including Thailand) [46,47]. However, Phoompanich et al., 2019 is the only study that applied ML for fire probability mapping in Thailand specifically, using Bayes Networks (BN) and Naïve Bayes (NB) to understand the interactions of fire with other hazards such as floods, landslides, and droughts [51].

While providing important information for wildfire management, these existing approaches for fire probability mapping face a few key challenges related to methods and data. With regard to methods, the MCA approaches are the simplest to implement, but scores and weights are completely subjective while lacking empirical measures of variable importance or standard measures of accuracy. Thus, final scores and accuracies can be difficult to compare to one another or interpret in real world scenarios [51,53–58]. Existing MCA and FDRS approaches have seen limited operational use by the general public in Thailand, due in part to how difficult they are to interpret in a practical setting without a fundamental understanding of the analysis methods [15]. ML approaches are computationally expensive and require large reference data sets for training, validation, and testing [59,60]. However, they can be used to empirically examine the relationship between predictor variables and fire occurrence and output probabilities that can be easily interpreted as the likelihood of a real world event [56,59,60]. In addition, ML based studies provide standard measures of accuracy and empirical measures of variable importances, allowing for inter-model comparison [54–56,58,60]. Despite ML increasingly becoming the standard for fire probability mapping [61], only one such study focuses specifically on Thailand and even relatively simple ML methods such as random forest have not yet been applied in a Thailand-specific context [47–52]. ML is also noticeably absent from Thailand's operational methods.

Existing approaches also do not include data-driven temporal variation between the predictor variables and predicted fire probabilities. Many of the approaches produce a single static map of fire probability representative of the entire time period of interest; these probabilities are based on cumulative fire occurrence over a set of years, and average conditions from the same years or a subset of those years [40,41,43–52]. They use historical fire reference points from a range of years, but these are aggregated into a single pool of binary data that disregards the date fires occurred; most predictor variables are static, and any historical predictor variables available for the time period are aggregated into a single layer for use in the model [40,41,43,45–52]. Thus, these models are not capable of associating predictor variables to reference points of each corresponding year to capture interannual variability of fire dynamics. Further, many studies derive their fire occurrence data solely from Moderate Resolution Imaging Spectroradiometer (MODIS) or Visible Infrared Imaging Radiometer Suite (VIIRS) active fire hotspots from NASA's Fire Information for Resource Management System (FIRMS) [45–47,51,62]. The inadequate spatial and

temporal resolution and sensor type make these datasets ineffective in detecting the small, short, and mostly anthropogenic fires common in Thailand.

This contrasts with other countries, where ML is becoming standard for operational methods and scientific literature is exploring more effective ways to account for temporal variability. In the United States, both large federal agencies and smaller regional organizations generate fire risk maps using ML models such as random forest and K-means clustering [63,64]. Studies from Italy, Spain, and the United States assess the implications of developing separate models for shorter time intervals such as weekends, months, seasons [65–67]. One study from Canada even explores the effectiveness of aggregating data over different time scales to test how influential interannual variability is compared to long-term averages [68].

To address both these methodological and data gaps in Thailand, we developed a seasonal fire probability mapping approach for Thailand, with a fully developed example from the nine northeastern provinces of the country. We leverage machine learning and globally available data sets in Google Earth Engine's (GEE) cloud computing environment. We trained a random forest model on annual historical fire and fire predictor data from 2016 to 2023 in Northern Thailand, then deployed the model on 2024 fire predictor data to generate the 2024 fire probability map. This methodology can increase the speed and complexity of fire risk related analyses while also offering flexibility for the input data sources and analysis frequency; it can be easily run for any given year and location for which historical fire data and predictor data are available.

Our intention was to develop a method to produce a straightforward probability metric that is easily interpreted by the general public and can be operationalized by government fire managers by running the analysis annually before each fire season. The purpose of this work is to provide an adaptable operational workflow, and with the explicit intention that agencies will substitute their own higher quality local data sets for the surrogate global datasets currently used in our model. In this light, our primary product is the analysis workflow itself, while the fire probability map is an interesting but auxiliary result. Our approach is novel in that we offer an operationalization-ready machine learning approach for fire probability mapping, improving upon the current operational methods in Thailand that use manual weighted overlay analyses; we use historical data to account for interannual variation in fire and its predictors in order to predict seasonal fire probability in a given year; we use a more representative sample design that allows for locations with any burn history to be selected as fire presences and absences for a given year; we empirically test and refine a comprehensive set of potential fire predictors compiled from existing fire probability mapping methodologies; and we provide all code necessary for implementation.

## 2. Materials and Methods

### 2.1. Study Area

The study area was the 9 most northwestern provinces of Thailand: Chiang Mai, Chiang Rai, Mae Hong Son, Lamphun, Lampang, Phayao, Phrae, Nan, Tak (Figure 1). These provinces experience lengthy fire seasons [2,42,51] characterized by infrequent, small, fragmented, and low intensity fires in forested and agricultural lands [2]. In 2019, the LULC comprised 69% closed forest (>70% canopy cover); 16% agriculture; 11% open forest (15–70% canopy cover); 2% urban; and <1% grassland, shrubland, or herbaceous wetland (<10% canopy cover) [69]. The terrain is mountainous, with some of the highest elevations found in Thailand [70]. The three main ecoregions are Kayah-Karen montane rain forests, Central Indochina dry forests, and Northern Thailand-Laos moist deciduous forests [71], all tropical and subtropical moist broadleaf forest biomes. The region is characterized by strong seasonality in precipitation and vegetation greenness; precipitation peaks in

September through October and the Normalized Difference Vegetation Index (NDVI) peaks in July through October [72–75]. Mean annual precipitation accumulation for 2016–2024 was 163 cm and mean temperature for 2016–2024 was 24 degrees Celsius [76]. Temperature generally fluctuates between 20–30 degrees Celsius throughout the year with the lowest temperatures in November through January and the highest temperatures in March through May [72].

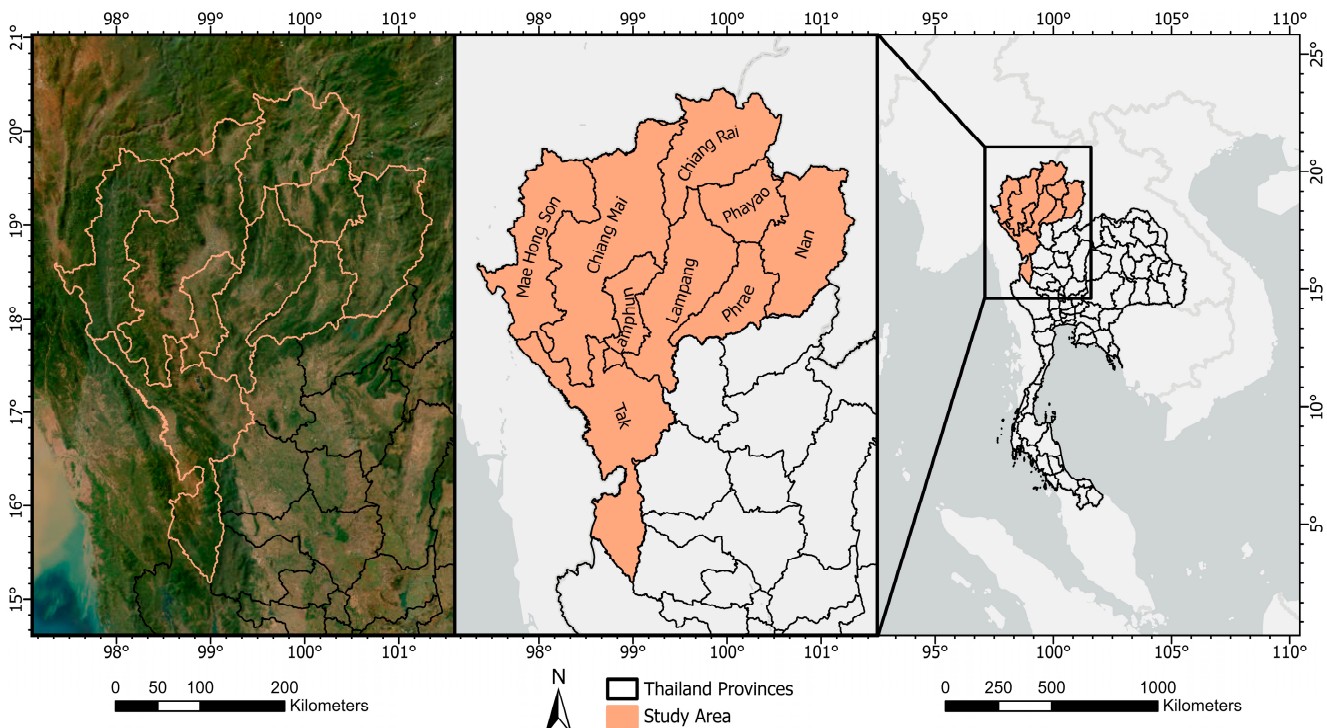

**Figure 1.** Reference maps of the study area, which consists of the 9 most northwestern provinces of Thailand. (Basemap credits: Earthstar Geographics, Esri, TomTom, Garmin, FAO, NOAA, USGS, © OpenStreetMap contributors, and the GIS User Community; Spatial reference: GCS WGS 1984, EPSG 4326).

*2.2. Model Selection*

We chose a random forest model to model fire probability due to both performance and operationalizability, both of which are central to operational use (Figure 2). We sought an algorithm with reliable performance with noisy multidimensional data, low barriers for implementation, and a native variable importance metric. Random forests are one of the only models that meet these criteria [77,78]. They perform well in predicting fire likelihood in Thailand and neighboring countries, sometimes even outperforming more complex machine learning models [47,48]. In this region, they have consistently achieved AUCs similar to or greater than those produced by Gradient Boosting Decision Trees (GBDT), Adaptive Boosting (AdaBoost), Support Vector Machines (SVM), and Particle Swarm Optimized Neural Fuzzy (PSO-NF) [47,48]. This pattern is consistent globally, where random forests either outperform other ML models or at least produce comparable results [61,78–89]. Random Forests are easy to operationalize, as they are conceptually intuitive algorithms that can be easily understood, employed, and refined by fire managers, even with no prior machine learning experience and no ongoing external technical support. They generally require less hyperparameter tuning for high performance than more complex methods [90]. Compared to simpler methods such as simple decision trees (DT), logistic regression (LR), Locally Weighted Learning (LWL), Bayes Networks (BN), and Naïve Bayes (NB), random forests better capture non-linear interactions between predictors [91] and are more robust

to overfitting due to data noise [92,93] and multicollinearity [94,95]. They also provide a built-in variable importance metric, which is absent or difficult to interpret in many other models [96–98]. Thus, random forests offer a compromise between simplicity, which lends itself to easier implementation, and complexity, which lends itself to more robust predictive power.

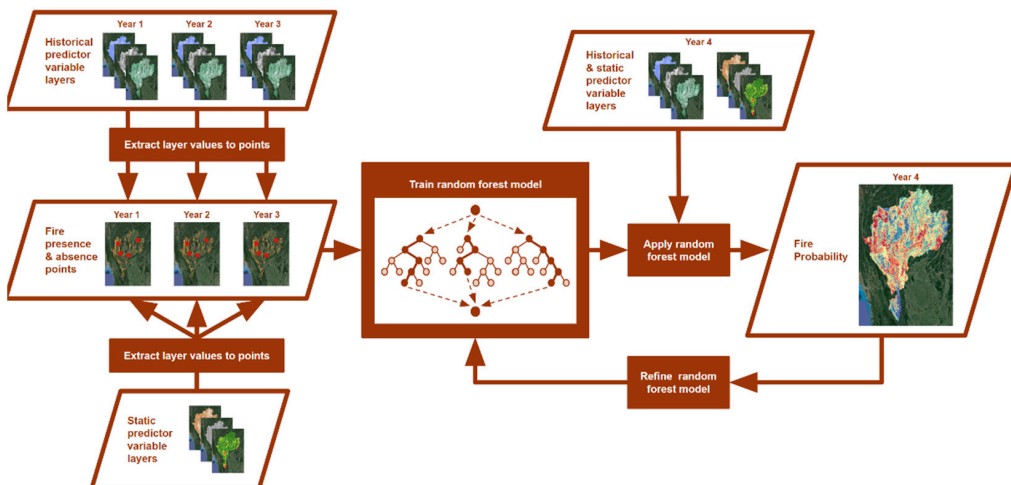

**Figure 2.** Overview of our model development workflow, including input data preparation, model development, and model refinement.

To create the seasonal fire probability map, we used Google Earth Engine's smileRandomForest classification algorithm with 500 decision trees [59,99]. Training points consisted of fire presence and absence data from 2016–2023, and environmental and social predictor variables consisted of representative data layers from 2016–2023. Once trained, the model was deployed on predictor variable layers from 2024 to produce the probability map for the 2024 fire season. The fire probability for each pixel was calculated as the percentage of the 500 decision trees that predicted fire presence at that pixel [59,100].

### 2.3. Data Selection

#### 2.3.1. Fire Presence and Absence Data

Reference points representing fire presence and absence were derived from burn scar polygons for 2014–2023 from GISTDA, which are the highest quality publicly accessible burned area data currently available for Thailand [1]. Each year, GISTDA comprehensively delineates all burned areas by calculating the difference between pre- and post-fire Normalized Burn Ratio (NBR). NBR is calculated from Sentinel-2 imagery to produce delta NBR (dNBR), which is then manually thresholded on a regional basis using its frequency distribution as a guide for determining the localized cutoffs for burnt areas [1,37,101]. GISTDA reports that the burn scar polygons have an overall accuracy of 66.8% [1], but in practice this accuracy is higher as our independent accuracy assessment yielded an overall accuracy of 83.3% (Supplement Table S4).

This approach is preferable to using MODIS or VIIRS-derived FIRMS active fire hotspots [62] as reference data for fire occurrence in Thailand for multiple reasons [102]. The MODIS and VIIRS sensors detect fires through temperature anomalies in their thermal bands, providing full global coverage once daily for MODIS and twice daily for VIIRS [5,103–105]. MODIS has a spatial resolution of 1000 m and VIIRS has a spatial resolution of 375 m [5,103]. MODIS-derived hotspots have low error of commission in Thailand [106] and VIIRS-derived hotspots can detect fires as small as 5 m² both globally [107] and in Thailand [108]. However, these data lack the spatial resolution necessary to pinpoint

precise locations for the small transient fires characteristic to Southeast Asia [102]. GISTDA burn scars in Thailand have a median size of 0.006 km$^2$ and a mean size of 0.15 km$^2$, compared to the 0.14–1 km$^2$ pixel size of the FIRMS data. FIRMS point locations are generated at the centroids of the MODIS or VIIRS pixels, so fires smaller than a pixel will have imprecise locations. Additionally, fires must produce enough heat to be detected during the satellite overpass [5,14,103]. Sensors sometimes do not register agricultural fires on small fields or understory fires under dense forest canopies, even if they burn during multiple satellite overpasses [15]. Further, landowners and officials are aware of satellite orbital patterns and will sometimes time their intentional burns to be between overpass times in order to avoid detection [15].

Reference points for fire presence and absence were generated with a stratified random sampling technique using the burn scar polygons from 2016–2023 obtained from GISTDA (Supplement Figure S2). Training and validation points (2016–2023) for model development were partitioned from the same pool of points, while testing points (2024) were generated separately from burn scar data received after the 2024 fire season was over. Points were stratified by both fire occurrence and year. For each year, we generated 300 points in areas which burned that year (fire presences) and 300 points in areas which did not burn that year (fire absences), with a minimum distance of 300 m between points; this yielded 600 points per year and 4800 points across all 8 years [109–111]. Because the GISTDA burn scar dataset is annually produced as a comprehensive map of all fires, the fire absence points can be presumed to be true absences for each year. Of these reference points, approximately 80% (3869) were used to train the model and approximately 20% (931) were used to validate the model during model refinement for 2016–2023; this yielded approximately 116 validation points per year, roughly evenly split between fire presence and absence [112]. An additional 600 reference points were generated solely for testing the final 2024 probability map (300 in areas that burned in 2024 and 300 in areas that did not burn in 2024).

### 2.3.2. Predictor Variable Data

We used environmental and socioeconomic variables with known relationships to fire probability in Thailand as predictor variables based on similar fire probability mapping analyses from scientific literature in peninsular Southeast Asia and operational methods in the Thai government [1,10,37,40–52]. These include variables representing topography [10,43,113,114], fuels [12,25,114–120], potential fire behavior [116,118], forest type [1,3,10,12,28,30,115,117,121–123], vegetation characteristics [3,12,28,30,115,117,121–123], climate [1,2,12,25,114,120,123], water availability [1,10,40,124], crop type [1,13,19,21–23], recent burn history [1,10,12,19,27,32,125], and human influence and accessibility [10,12,123] (Table 1). These variables both directly and indirectly influence ignition likelihood and fire behavior, and many of them exhibit complex interactions with each other across multiple temporal and spatial scales [19,28,113,117,121,122,126–129].

**Table 1.** Predictor variables included in the full model prior to model refinement.

| Environmental Variables | | | | | | |
|---|---|---|---|---|---|---|
| Category | Variable | Units | Description | Available Temporal Extent | Available Spatial Extent | Data Source |
| Topography | Elevation | meters | Elevation above sea level | NA | global | [70] |
| | Slope | degrees | Degree of incline | NA | global | [70] |
| | Aspect | degrees | Orientation of slope | NA | global | [70] |

<div align="center">**Table 1.** *Cont.*</div>

**Environmental Variables**

| Category | Variable | Units | Description | Available Temporal Extent | Available Spatial Extent | Data Source |
|---|---|---|---|---|---|---|
| Fuels | Woody and herbaceous fuel load | tons per ha | Combined mass of fuel from sound woody and primary herbaceous vegetation | 2015 | global | [130,131] |
| | Litter cover | percent | Percent of ground cover of leaf litter | 2015 | global | [130] |
| | Litter depth | centimeters | Depth of vegetative litter | 2015 | global | [130] |
| | Grass height | centimeters | Height of primary herbaceous vegetation | 2015 | global | [130] |
| Potential fire behavior | Flame length | meters | Modeled flame length from the Fuel Characteristic Classification System | 2015 | global | [118,130–133] |
| | Rate of fire spread | meters per minute | Modeled rate of fire spread from the Fuel Characteristic Classification System | 2015 | global | [116,130,131,133] |
| Forest type | Distance to forest type | kilometers | Distance to common forest types<br>1. Dry Evergreen Forest<br>2. Hill Evergreen Forest<br>3. Pine Forest<br>4. Mixed Deciduous Forest<br>5. Dry Dipterocarp Forest<br>6. Bamboo Forest<br>7. Teak Plantation<br>8. Secondary Growth Forest<br>9. Old clearing<br>10. Eucalyptus Plantation | NA | Thailand | RFD * |
| Vegetation Characteristics | Canopy cover | percent | Percent of cover of trees from above (peak of growing season) | 2000–2023, annual | Mekong region | [134] ** |
| | Change in canopy cover | percent | Difference in canopy cover between current year and prior year; positive values indicate increase, negative values indicate decrease | 2001–2023, annual | Mekong region | [134] ** |
| | Change in canopy height | meters | Difference in canopy height between current year and prior year; positive values indicate increase, negative values indicate decrease | 2001–2023, annual | Mekong region | [135] ** |
| | Normalized difference moisture index (NDMI) | unitless | Captures moisture content of vegetation; calculated from the near-infrared and shortwave infrared bands using the formula (SWIR2-Red)/(SWIR2+Red); positive values indicate higher moisture, negative values indicate lower moisture | 2013–2024 | global | [136] |
| | Enhanced vegetation index (EVI) | unitless | Captures density and health of vegetation; calculated from the red, blue, and near-infrared bands using the formula $2.5 \times$ (NIR-red)/(NIR + ($6 \times$ red) − ($7.5 \times$ blue) + 1); positive values indicate higher moisture, negative values indicate lower moisture | 2013–2024 | global | [136] |
| | Seasonal Differences in HH SAR signal | decibels | Difference between Synthetic Aperture Radar (SAR) HH polarization backscatter between the wet and dry seasons | 2014–2024 | global | [137] |

**Table 1.** *Cont.*

**Environmental Variables**

| Category | Variable | Units | Description | Available Temporal Extent | Available Spatial Extent | Data Source |
|---|---|---|---|---|---|---|
| Climate | Maximum temperature | degrees Celsius | Average maximum temperature of air at 2 m above the earth surface | 1958–2023, monthly | global | [138,139] |
| | Precipitation | millimeters | Sum of accumulated precipitation | 1958–2023, monthly | global | [138,139] |
| | Vapor pressure deficit (VPD) | kilopascals | Difference between the amount of moisture in the air and how much moisture the air can hold when it is saturated; calculated from dewpoint temperature and temperature | 1958–2023, monthly | global | [138,139] |
| | Soil moisture | millimeters | Water content of soil; calculated using a one-dimensional soil water balance model | 1958–2023, monthly | global | [138,139] |
| | Palmer drought severity index (PDSI) | unitless | Quantifies long-term drought and can be interpreted as relative dryness as a deviation from normal conditions; calculated from temperature data and precipitation data with a physical water balance model; values range from $-10$ to 10, negative values indicate dryer conditions and positive values indicate wetter conditions | 1958–2023, monthly | global | [138,139] |
| Water Availability | Distance to water | kilometers | Distance to natural and artificial sources of water (Farm ponds, Irrigation canals, Oceans, Reservoirs, Lakes, Lagoons, Rivers, Canals) | NA | Thailand | LDD *** |
| | Normalized difference water index (NDWI) | unitless | Captures the presence of open water bodies and moisture content of vegetation; calculated from the red and near-infrared bands; positive values indicate surface water present, negative values indicate no surface water present | 2013–2024 | global | [136] |

**Socioeconomic Variables**

| Category | Variable | Units | Description | Available Temporal Extent | Available Spatial Extent | Data Source |
|---|---|---|---|---|---|---|
| Crop type | Distance to crop types | kilometers | Distance to crop types managed with fire 1.Maize 2. Corn 3. Sugarcane | NA | Thailand | LDD *** |
| Recent burn history | Distance to burns (1 & 2 years prior) | kilometers | Distance to burn scars that occurred 1 year prior to the year of interest and 2 years prior to the year of interest | 2015–2023, annual | Thailand | GISTDA **** |

**Table 1.** *Cont.*

| Socioeconomic Variables | | | | | | |
|---|---|---|---|---|---|---|
| **Category** | **Variable** | **Units** | **Description** | **Available Temporal Extent** | **Available Spatial Extent** | **Data Source** |
| Human influence and accessibility | Distance to roads | kilometers | Distance to roads | NA | Thailand | [140] |
| | Distance to settlements | kilometers | Distance to buildings | 2015 | global | [141,142] |
| | Distance to SPK & KTC areas | kilometers | Distance to land under special agricultural management provisions in either the Sor Por Kor (SPK) and Kor Tor Chor (KTC) programs (land reform areas under laws M64 and M121) | NA | Thailand | [143] |
| | Distance to DNP & RFD areas | kilometers | Distance to land under the jurisdiction and protection of either the Department of National Parks (DNP) or the Royal Forestry Department (RFD) | NA | Thailand | [144,145] |
| | Population count | people per hectare | Population density, represented by the number of people residing per hectare | 1975–2030, 5-year intervals | global | [146,147] |

* Provided directly by the Royal Forest Department (RFD); ** A regional version of the referenced model, provided directly by University of Maryland; *** Provided directly by the Land Development Department (LDD); **** Provided directly by the Geo-Informatics & Space Technology Development Agency (GISDTA).

### 2.4. Data Preprocessing

Full descriptions of preprocessing steps and example maps for each predictor layer can be found in Supplement Table S3 and Supplement Figure S10. Publicly available global data layers were used to represent predictors for which we did not have local data sets. All predictor variable layers were resampled to 300 m, the resolution of the coarsest resolution terrestrial data set, using either mean or minimum aggregating functions as appropriate. For the climate data sets, which had spatial resolutions lower than 300 m (pixel size larger than 300 m), the pixels were subdivided to 300 m and smoothed with a focal mean function. This approach served as a compromise between the wide range of resolutions in the available data sets, with the goal of maintaining fine-scale ecological variation in the sub-500 m data sets while minimizing edge artifacts and spurious precision introduced by down sampling in the 4+ km data sets.

All variables with multiple time steps available were composited to produce operational layers comprising data available before the start of each fire season. The exact compositing methods varied based on the known seasonality of the variables and the data type, quality, and availability. For example, climate variables were derived from composites of the 5 pre-fire season months (August–December) [1–4], forest canopy and change variables were derived from annual values from the two pre-fire season years, optical imagery indices were derived from composites of the pre-fire season year, and seasonal SAR imagery differences were derived from composites of the 4 months in the peak leaf-on season (July–October) and 4 months in peak leaf-off season (January–April) [72–75]. For data sets that had temporal resolutions lower than 1 year (intervals greater than 1 year), annual layers for years without data were created using the most recent values. Predictor variable values were extracted to the reference points from the corresponding year. For predictors for which we had no historical data, the values from the single-date static layers were extracted to all points.

In order to ensure the reliability of variable importance assessments, we ensured that all variables were continuous and scaled to the same numerical range. The Gini Index favors variables that provide more potential splits during tree construction, including variables with more unique values and larger numerical ranges [97,148]. Thus, it often assigns higher importance to continuous rather than categorical variables, and higher importance to variables with higher numerical values rather than lower numerical values [97,148]. Additionally, if some rare classes have very few representative reference points, the model may not be able to identify their predictive relationship with fire, even if a relationship does exist [149]. Categorical variables were converted to continuous values by calculating the distance to each class, alleviating the biases posed by categorical variables and sampling bias. All predictor variable values were also normalized to a zero to one scale using a simple min-max normalization, alleviating the biases posed by differing numerical ranges.

### 2.5. Model Development and Refinement

#### 2.5.1. Multicollinearity

We assessed multicollinearity between variables using correlation coefficients and the Variance Inflation Factor (VIF) [150,151] (Supplement Figures S3–S5). While random forest models have been shown to maintain predictive performance despite multicollinearity in predictor variables [94,95], multicollinearity can confound variable importance rankings in random forest models [149,152,153]. Two pairs of variables had high correlation, namely elevation and maximum temperature and EVI and NDWI, however we kept both pairs as they independently influence fire behavior and their VIF fell below the generally accepted threshold [151]. Ecological data often exhibit inherent multicollinearity, and it can be appropriate to retain highly correlated variables in random forest models when the goal of the study includes identifying multiple important variables [149].

#### 2.5.2. Variable Importance

Variable importance was calculated for all predictors using the Gini Index. Despite its potential biases, Gini impurity is the most accessible empirical variable importance metric on which to base variable selection for an operational fire probability model. The Gini Index measures how often a randomly chosen data point would be incorrectly classified if it were given a random class label based on the distribution of classes in a given subset of data points [96,154]. The variable importance metric can be defined as how much each variable contributes to the model's ability to distinguish between classes.

We normalized variable importances to sum to a 100 for easy interpretation. The model was then refined by removing variables individually in order of importance to the model until further removals resulted in a distinct drop in 2016–2023 AUC below 0.84 (Figure 3). Eleven variables were removed during model refinement, leaving 31 variables in the final version of the model. A full chart of variable importances before model refinement can be found in Supplement Figure S6.

#### 2.5.3. Accuracy Assessment

We used Area Under the Curve (AUC), which is calculated for the Receiver Operating Characteristic (ROC) curve, to assess the predictive power of the model [47–50,52,155–158]. AUC quantifies a model's ability to correctly classify the validation and testing data compared to a random classification; values between 0.8 and 0.9 are excellent and values between 0.9 and 1.0 are outstanding [159–161]. We calculated AUC for the two separate data sets: validation points from 2016–2023 (the approximately 116 yearly points generated for 2016–2023, which yielded a total of 931 points or approximately 20% of the initial 4800 points) and testing points from 2024 (the 600 additional points generated for 2024) [112]. Further details about reference data generation and partitioning can be found in Methods

Section 2.3.1. To provide insight into the spatial distribution of model uncertainty, we generated a map of binomial standard error at each pixel, treating the proportion of trees voting for fire as a binomial variable (Supplement Figure S7) [162].

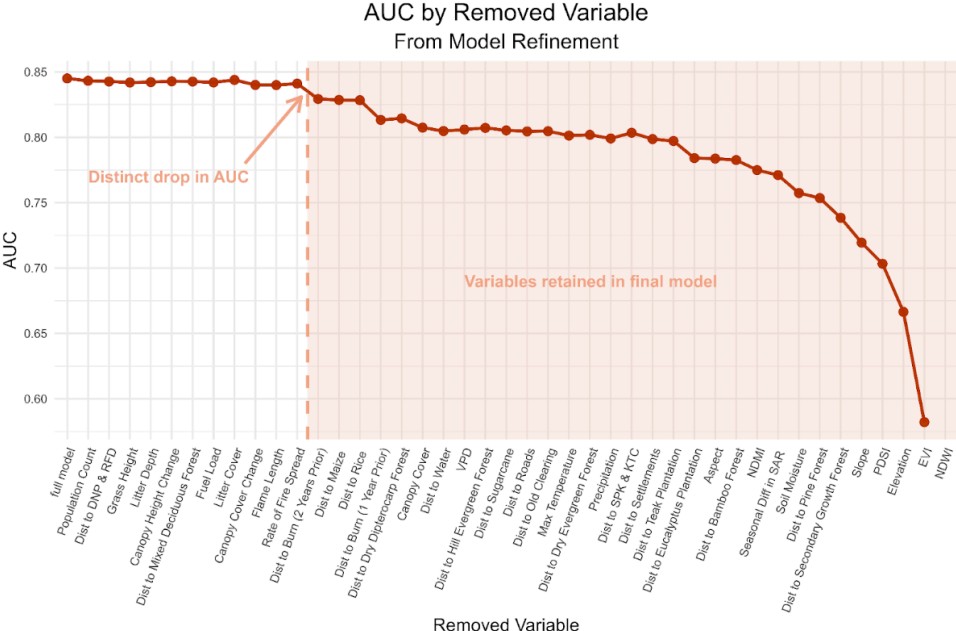

**Figure 3.** 2016–2023 AUC of all model runs during model refinement using consecutive variable removal without replacement.

2.5.4. Sensitivity Analysis

We also conducted a sensitivity analysis on the model's response to the removal of individual variables. We trained and validated 44 versions of the random forest model, with the initial run including all variables and each subsequent run removing only a single variable from the set. We calculated the mean and median of each variable's scaled importance values across all runs and plotted the difference in 2016–2023 AUC (dAUC) between each run and the initial run.

## 3. Results

### 3.1. Variable Importance and Model Sensitivity

Two distinct groups emerged within our variable importance rankings: a group of low importance variables, and a group of high importance variables with nearly equal importance values. There is a clear difference in importance values between the top and bottom groups and a distinct drop in 2016–2023 AUC once variables from the high importance group are removed during model refinement (Figure 3). Before model refinement, the most important variables each contributed between 2.4–3.1% of the total impurity reduction across all splits within all trees (Supplement Figure S6). These included certain forest and crop types, vegetation characteristics, topography, climate, human influence and accessibility, water availability, and recent burn history (Figure 4).

In the sensitivity analysis, seasonal difference in SAR backscatter, PDSI, aspect, distance to bamboo forest, and EVI had the highest mean and median importance values across all runs, while population density, distance to Royal Forest Department (RFD) and Department of National Parks, Wildlife, and Plant Conservation (DNP) protected areas, grass height, litter depth, and canopy height change had the lowest importance values (Figure 5). Removal of distance to hill evergreen forest, soil moisture, canopy height change, and slope increased the 2016–2023 AUC, while the removal of distance to burns

2 years prior, distance to burns 1 year prior, and canopy cover decreased the 2016–2023 AUC (Figure 6).

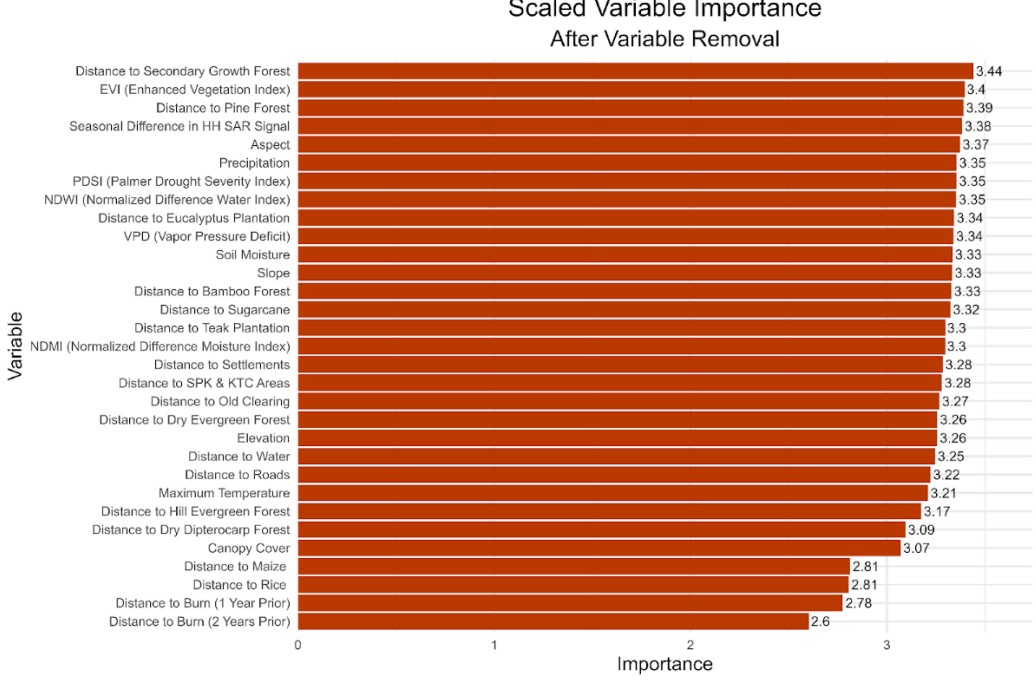

**Figure 4.** Scaled Gini-based variable importance values of final model, after model refinement through variable removal.

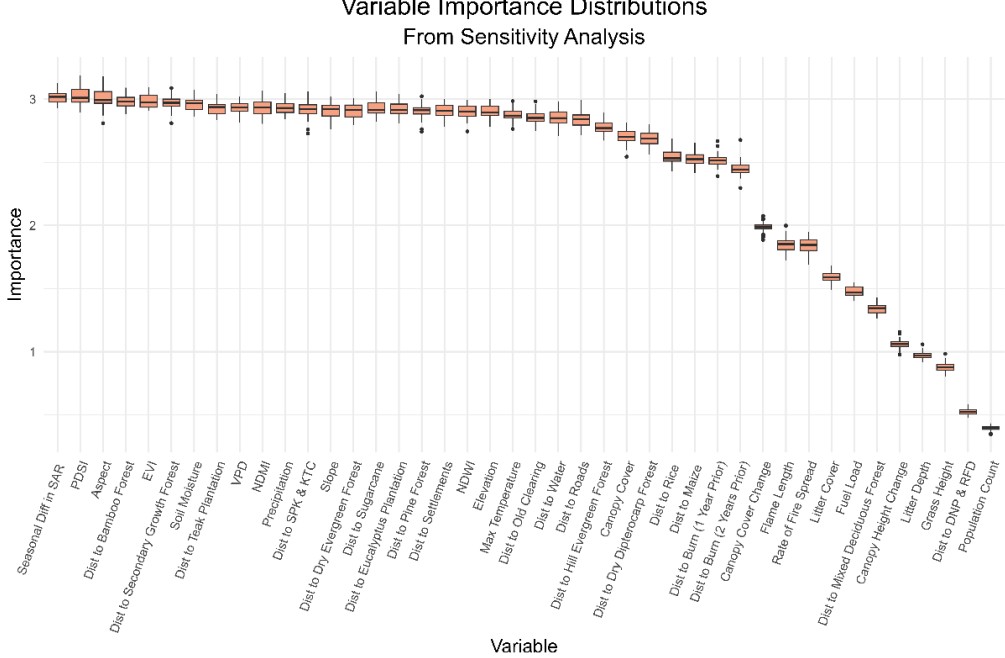

**Figure 5.** Variable importance distributions of predictor variables across all model runs in sensitivity analysis using individual variable removal with replacement.

### 3.2. Fire Probability Map

The 2024 Fire Probability in Northern Thailand map (Figure 7) is available online as an interactive application in Google Earth Engine (https://worldbank-fire.projects. earthengine.app/view/fire-probability-thailand, accessed on 2 September 2025).

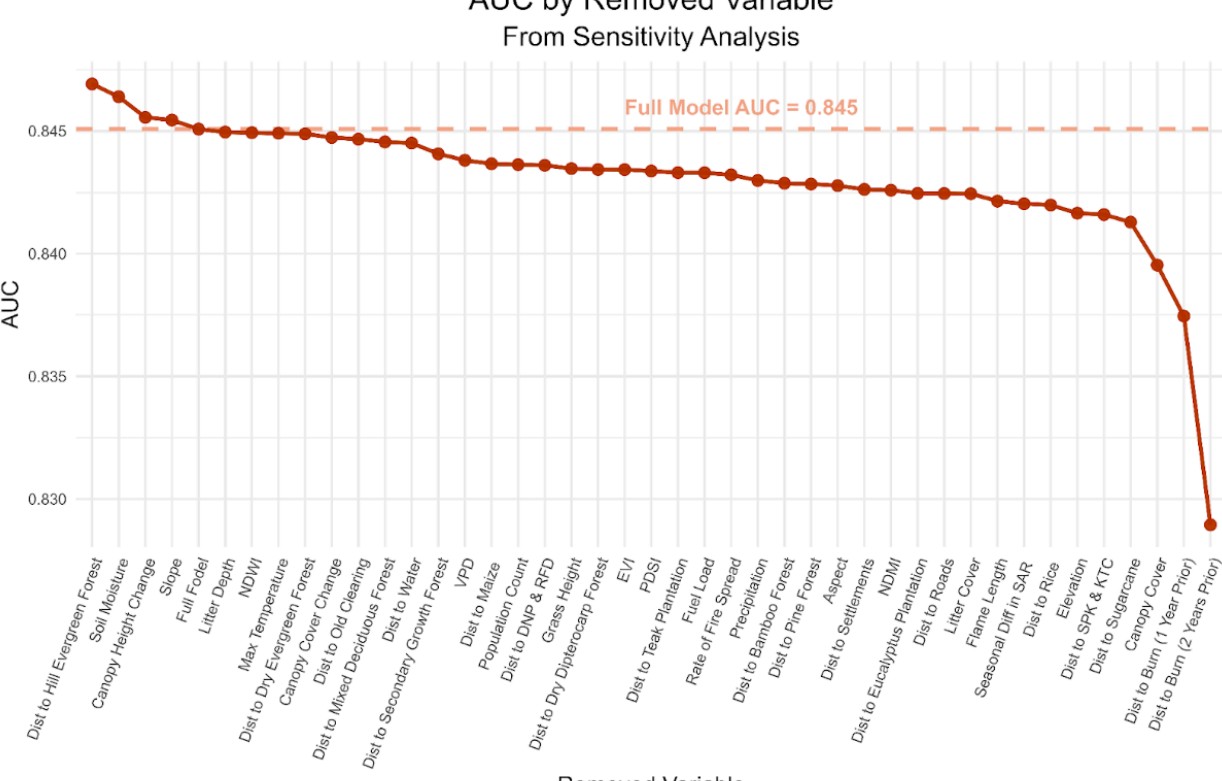

**Figure 6.** 2016–2023 AUC of all model runs in sensitivity analysis using individual variable removal with replacement.

The overall AUC of our fire probability model is 0.841 for 2016–2023 and 0.848 for 2024, which demonstrates a reasonably high ability to discriminate between fire presence and absence [159–161]. The ROC curves show that as the classification threshold is gradually lowered, the true positive rate increases rapidly but the false positive rate increases more slowly (Supplement Figure S8). This indicates that the model can distinguish well between areas that have burned and have not burned using the selected predictor variables in the area and time period of interest. Our model's AUC is comparable to the AUCs obtained from similar ML models in Southeast Asia, which range from 0.81 for He et al. to Adaptive Boosting in 2021 of 0.98 for Tuyen et al. and Locally Weighted Learning with Dagging in 2021 [49,50,52]. When comparing specifically to the two random forest models in this region, our model's AUC is slightly lower than those obtained by He et al., 2021 (AUC of 0.91) and Tien Bui et al., 2017 (AUC of 0.906) [47,48]. The reasons for this are explored further in Discussion Section 4.1. Our model's accuracy is difficult to quantitatively compare to those of MCA methods, as those approaches rely on confusion-matrix–based metrics derived from testing points grouped into subjective ordinal categories, which are not directly comparable to probability-based ML performance metrics like AUC.

The prediction map's standard errors (Supplement Figure S7) reflect that areas with extremely high and extremely low fire probability have the highest degree of certainty. Pixels with high consensus among decision trees have a consistent combination of important predictor variables correlated with either the presence or absence of fire.

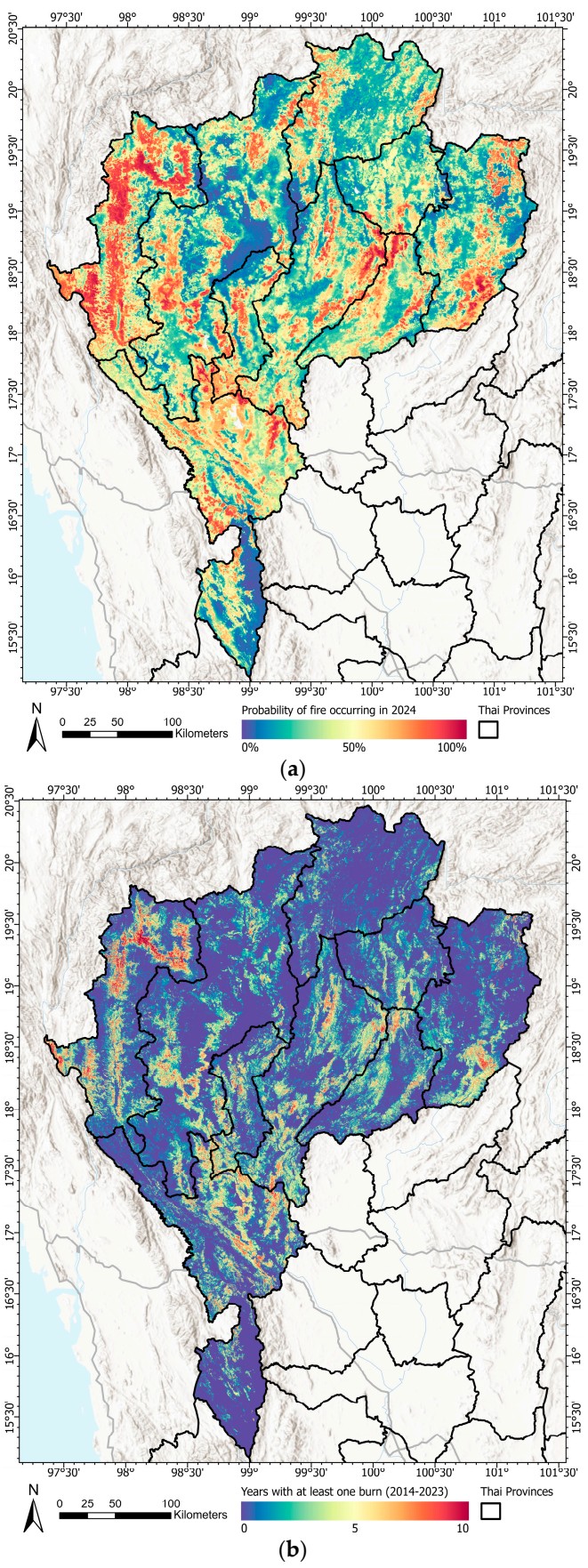

**Figure 7.** *Cont.*

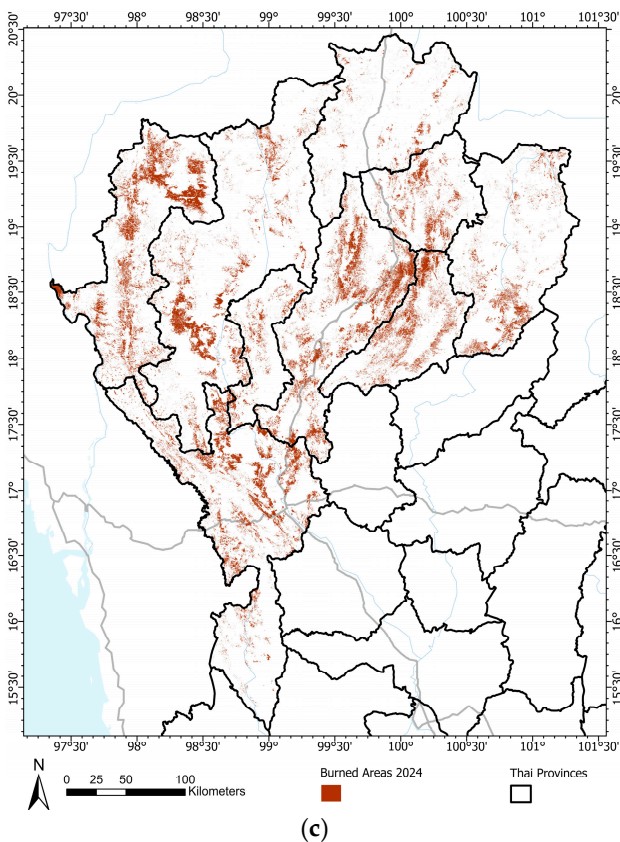

(**c**)

**Figure 7.** (**a**) The probability of fire occurrence in 2024 for the 9 most northwestern provinces of Thailand, generated by the random forest model developed with predictor and fire occurrence data from 2016–2023 and applied to new predictor data from 2024; (**b**) Fire frequency (number of years with at least one fire occurrence) for the 9 most northwestern provinces of Thailand from 2014–2023; (**c**) True burned areas in 2024. (Basemap credits: Esri, CGIAR, USGS, TomTom, Garmin, FAO, NOAA, USGS, © OpenStreetMap contributors, and the GIS User Community; Spatial reference: GCS WGS 1984, EPSG 4326).

## 4. Discussion

### 4.1. Evaluation of Model Results

Most fires in Thailand are anthropogenic and ignited for land management purposes in agricultural production or non-timber forest product collection [1,12,13,15–19]. Our 2024 fire probability map aligns well with these known patterns, indicating that the highest probability of fire is at mid-elevations in agriculture and dry, deciduous, or disturbed forests, near and accessible to human settlements by roads but not directly adjacent to them. The lowest probability of fire is in dense urban areas, moist evergreen forests, and lowland wet rice fields. There are also distinct regions with moderate fire probability, indicating high model uncertainty; these include cropland and forest that is only burned on an irregular multi-year schedule or only occasionally experiences escaped wildfires. Empirically, our model also performed well, with an AUC of 0.841 for the 2016–2023 validation points and an AUC of 0.848 for the 2024 testing points [159–161].

Overall, despite two major differences in methods, our seasonal spatial patterns align with other machine learning based fire probability approaches in Thailand and SE Asia. First, while other studies used a small number of predictor variables based on theory [47–52], we began with a large number of predictor variables and iteratively removed empirically less important variables. We found that certain forest and crop types, vegetation characteristics (structure, seasonality, health, density), topography (elevation,

aspect, slope), climate (precipitation, temperature, VPD, soil moisture, drought indicators), human influence and accessibility (roads, settlements, special management designations), water availability, and recent burn history are the strongest predictors of fire probability in northern Thailand (Figure 4, Supplement Figure S6). These variables were all of near equal importance in our model, and our sensitivity analysis confirmed the decision to retain these variables (Figures 5 and 6). This is in contrast with similar studies that found only a few variables of high importance, with the highest often being distance to roads and human settlements [47–50,52].

As with the spatial distribution of fire probability, these variable importances reinforce the known drivers of fire in Thailand. Topography, climate, and water availability are universally understood as foundational components of fire prediction [163,164], while human influence and accessibility, recent burn history, vegetation characteristics, and forest and crop type are important for reasons unique to the ecological and socioeconomic systems of Thailand. Because most fires originate from intentional burning, human accessibility in terms of proximity to infrastructure is an important constraint for where fires occur. Once a plot of land is initially burned, it is likely to be burned again for continued production in the coming years, making recent burn history a likewise useful predictive variable. Maize, rice, and sugarcane are the most commonly burned crops, making these fields frequent fire ignition points and spread pathways. In forests, tree species, structure, seasonality, health, density are critical factors for fire ignition and spread, both because humans utilize certain species and communities more and because certain ecosystems are inherently more fire prone. For example, drier deciduous or coniferous forest types, particularly those with savanna-like characteristics, generate more flammable fuel, with many exhibiting fire adapted traits and long histories of natural fire regimes. Plantation forests such as bamboo, teak, and eucalyptus, as well as naturally occurring dry dipterocarp forests, are both flammable and intentionally burned. Meanwhile, secondary forest types with recent disturbance are likely still managed or situated near managed land, in close proximity to intentional burns.

Differences in variable importance may be attributed to differences in model structure, input data quality, and variable importance metrics, complicating direct comparisons. Specifically, other studies use fewer variables with minimal multicollinearity (e.g., [47]), or different variable importance metrics (e.g., Relief-F Average Merit [49], Correlation Attribute Evaluation [50], Pearson Correlation Coefficient [48,52], or Cramer's V Coefficient [46]). Some of our observed patterns in variable importance may also stem from the statistical properties of the Gini Index, namely its sensitivity to high numbers of variables [149], sampling bias in variables [149], multicollinearity among variables [149,152,153], and differing unique value counts across variables [97,148,152]. This may have contributed to the low importance we observed for population density, distance to protected areas, fuel characteristics, predicted fire behavior, and canopy change, along with the discrepancies between dAUCs and Gini Index rankings in our sensitivity analysis.

Further, many of these metrics, including the Gini Index, cannot capture spatial autocorrelation [96,165], which is important for many geospatial phenomena including fire. Future research should evaluate variable importance metrics in the context of fire probability mapping in Thailand in order to understand the practical implications and tradeoffs of using different metrics to build machine learning models for management applications. We also suggest including forecasts for fire season weather conditions; these weather conditions influence fire behavior in real time and have a different effect than pre-fire season climate which largely influences fuel conditions. Likewise, we recommend incorporating a longer history of multi-year climatological phenomena, both cyclical like ENSO and sporadic like droughts, to capture their lag effects on ecosystems and agriculture.

Including crop rotation and shifting cultivation as predictor variables may also improve the model; agriculture is particularly transient in nature, as different crop types are often rotated seasonally on a single plot of land and forests are frequently cleared to be cultivated or grazed for only a few years. The model can also be applied with future climate projections as the input climate predictor variables to see how different climate change scenarios will alter fire probability. An important component of this is also calculating the uncertainty associated with each set of conditions.

The second major methodological difference for our approach is that we paired annual fire occurrences with annual predictor variables in order to capture interannual temporal variation, while other studies use cumulative fire occurrence with temporally averaged predictor conditions [47–52]. This allows us to evaluate year-to-year shifts following monotonic (deforestation, climate change), sporadic (storms, land ownership transitions), and cyclical (ENSO, crop cycles) trends. In contrast, when interannual variations in predictor conditions are aggregated for the study period, their influence on fire probability is also aggregated; thus, cyclical patterns during the study period may be discounted. Our temporal disaggregation also highlights a stark spatial divide between areas with high versus moderate fire probabilities, a difference that is likely because some areas burn nearly every year while others burn only intermittently. The cultural customs and economic pressures that drive these fire intervals can be highly localized [15]. This has important management implications for the decisionmakers tasked with prioritizing resources for the fire season, and it is critical to track them more closely.

This approach has important implications for both the model training data selection and calculating accuracy metrics. Considering temporal variation allowed us to draw a more representative sample for fire presence and absence. Other approaches define fire absences as locations that never burned within a multi-year study period [47–52], but our model also considers areas that may have burned in one year in the study period but not in other years as fire absences for those years in which the area did not burn (Supplement Figure S9). This allowed us to draw a more representative sample for fire absences, as there are fundamental ecological and socioeconomic differences between areas that have never burned in the study period and areas that have burned, but not during the specific year of interest. Sampling fire absences only from areas that never burned creates a biased sample that provides an extreme basis for comparison with fire presences. Further, with each new fire season, the model can be retrained and rerefined with the addition of the most recent fire occurrence and fire predictor data to enhance its performance.

The increased temporal resolution and representative sample design likely contributed to the slightly lower AUC compared to the $\geq 0.9$ values reported by other ML models [47–50,52]. Higher temporal resolution preserves temporal variability—e.g., in climate and agricultural cycles—that would otherwise be smoothed by multi-year averages. Including areas that burned during the study period but not in a specific year better represents the fire regime in the model but slightly reduces predictive power; this is also underpinned by the fact that the areas of lowest model certainty are found in intermittently burned cropland and forest. In contrast, other models often use a biased subset of fire absences and smooth inter-annual variability, potentially exaggerating predictor-response relationships and inflating AUCs. While most models capture only spatial variation, ours accounts for both spatial and temporal variability, introducing greater complexity and noise due to the more representative sampling and added temporal dimension. Thus, the AUCs of other models may have been artificially inflated by biases in the sample designs, some of which are mitigated in our approach. These differences suggest future work should examine how model performance varies with the selection of training, validation, and testing years to optimize prediction accuracy while minimizing data requirements (Supplement

Table S5). Further, we suggest systematically testing a broader range of ML approaches against each other with our data to determine their strengths and weaknesses, contextualizing these results with those of other comparative ML studies from across Southeast Asia (Supplement Table S2).

### 4.2. Feedback from Stakeholders

Stakeholders provided feedback on a preliminary version of the fire probability mapping approach at a Regional Consultation in Chiang Mai Province and National Consultation in Bangkok. The proposed methods were well received, especially by technical geospatial experts who are already familiar with the advantages of machine learning and cloud computing and are interested in expanding their departments' use of these tools. However, despite the relative simplicity of these methods compared with other approaches in the global literature, many participants also recognized the organizational and technical obstacles that would need to be addressed in order to implement these methods [59,60]. This would include widespread capacity building and data collection, as well as changes in organizational structure to integrate automated tools with manual methodologies.

Stakeholders also expressed the need to identify the intended uses and users of the fire probability map, from high level officials and local communities. In its current form, the map can provide a broad overview of expected fire patterns, but it should be further calibrated for community use. Community-level data should be incorporated to refine the analysis, since local communities are one of the primary actors in preventing and fighting fires [15]. Local communities would need much greater spatial and temporal precision in the map for it to be useful for on-the-ground activities, which can be achieved if such products with the desired spatial and temporal resolution are incorporated into the model in lieu of the global data sets currently used as placeholders [15].

These concerns about data quality were also reflected in the final model's accuracy metrics. The reference data points were produced from the burn scar polygons provided by GISTDA, which were themselves generated using an automated methodology. These data are likely less accurate than data produced by manual image interpretation or field surveys (supplement Table S4). Similarly, the fuel, potential fire behavior, climate, and canopy data layers were derived from global data sets, which were generated using generalized models at the global or continental scales [130,134,135,138]. These data are likely less precise than regional or national scale data that are generated using locally calibrated and validated models.

These concerns can be addressed by replacing surrogate data sets with improved data from local experts. Due to limited data sharing and accessibility, our current model uses coarse, global public datasets for many predictors like climate, weather, and fuels, producing a fire probability map with lower resolution than the scale of fires in Thailand. For variables like LULC and infrastructure, only static, single-date datasets were available, which miss rapid interannual changes that may affect fire probability. Replacing them with high-resolution, historical local data would improve model performance and operational value. Some datasets, like climate and weather, are regularly produced by the government but remain hard to access, while others, like fuels and corresponding fire behavior, do not yet exist and would require extensive fieldwork and modeling to produce. We also recommend generating reference data from manually delineated burn scars to improve training and testing accuracy. Likewise, localization efforts should incorporate data layers representing the spatiotemporal patterns of local land use customs, such as crop-specific burn cycles.

Further, more granular stratification of reference points based on LULC type could minimize sampling bias and uncover sources of spatial autocorrelation not addressed

in our current sample design. However, it is important to note that spatial autocorrelation is an inherent property of fire occurrence that can be informative of underlying processes [1,12,13,15–19]. The fundamental mechanisms of fire regimes, their drivers, and human management differ substantially between LULC classes [1,10,12,13,15–24,37,40–52]. Thus, developing separate fire probability models for each LULC type could help further isolate meaningful sources of spatial autocorrelation, including broad differences between classes and finer-scale variation within them. Since we transformed the categorical LULC data to continuous distance layers to minimize biases in the Gini-based variable importance rankings, this complicates the interpretation of the relationships between LULC, fire, and the other predictor variables. Therefore, building and comparing separate models for key LULC types, such as specific crops or forest types, is advisable.

Finally, local users would benefit from higher temporal resolution in fire probability maps. Shifting from seasonal to daily, weekly, or monthly predictions would improve prevention, mitigation, and response during fire season. Weather, a key driver of fire behavior, changes frequently across these timescales. Moreover, intentional burning follows irregular but somewhat predictable schedules for each type of crop or forest, depending on its unique cultivation needs, site characteristics, and cultural norms [15]. More frequent predictions would be possible if the necessary temporal data were available for these key variables. This includes date-labeled fire occurrences, short-term weather forecasts, and detailed regional calendars of burning activities that temporally align with the desired prediction intervals.

## 5. Conclusions

With machine learning and other forms of artificial intelligence becoming standard tools in the field of fire probability mapping worldwide [61,78,166–168], Thailand should leverage these tools for their own fire management activities. Our approach provides the groundwork for an easy-to implement approach that predicts fire probability annually and can be adapted based on locally available data.

We present an annual random forest machine learning model approach to mapping fire probability in northern Thailand that produces a straightforward fire probability metric. The model yielded a high AUC of 0.841 for 2016–2023 and 0.848 for 2024, and indicated the highest probability of fire is in easily accessed agricultural and forested areas adjacent to human settlements. Variables with high predictive power in the model included certain forest and crop types, vegetation characteristics, topography, climate, human influence and accessibility, water availability, and recent burn history. The model aligns with both previous maps of fire probability and known mechanistic drivers of fire in Thailand.

Key process improvements include empirical variable selection, disaggregated historical data, and representative sample design. Systematically selecting the most influential fire predictor variables from a comprehensive set of predictor variables, rather than pre-selecting variables, allows flexibility and opportunities for localization. Disaggregating historical data to account for year-to-year variability in fire predictors makes the model sensitive to cyclical and sporadic variables. This, in turn, allows for a more representative sample for fire absences, including areas that did not burn in a given year rather than only areas that never burned.

Thus, our methodology sets up a framework for the future scaling of the analysis, both in the spatial and temporal dimensions. Due to the modularity and automation employed by the model, it can be modified using higher quality local data in lieu of global datasets for localization. This makes it adaptable for data scarce regions, using fewer years of fire occurrences or predictor variables until more data can be made available. Empirical variable selection allows for spatial localization, where individual models can be tuned

to unique fire dynamics within different LULC classes. Disaggregation of historical data allows for increased temporal cadence, where predictor variables can be aggregated on smaller time scales to produce monthly, weekly, or daily maps throughout the fire season.

Our workflow provides an empirical, quantitative approach for predicting wildfire probability based on the specific conditions leading up to each fire season. Knowing where to expect fire is critical for implementing prevention measures and allocating response resources before fires actually occur. Our model provides both reliable predictive performance and straightforward interpretability, making it an effective decision support tool that could advance fire management in Thailand.

**Supplementary Materials:** The following supporting information can be downloaded at: https://www.mdpi.com/article/10.3390/rs17193378/s1, Figure S1: Causes of fire and the feedback mechanisms that drive wildfires in Thailand; after fire is initially introduced to a landscape, socioeconomic and environmental feedbacks may increase the likelihood of future fires, Figure S2: Reference points used to train and validate the model, visualized by (Left) fire absence and presence and (Right) year, Figure S3: Strongest correlations among predictor variables (R2 > 0.5), prior to model refinement through variable removal, Figure S4: Variance Inflation Factors (VIF) of predictor variables, prior to model refinement through variable removal, Figure S5: All correlations among predictor variables, prior to model refinement through variable removal, Figure S6: Scaled Gini-based variable importance values of the initial model, prior to model refinement through variable removal, Figure S7: The binomial standard error of the 2024 fire probability map, Figure S8: Receiver Operating Characteristic (ROC) curve of the final model (after model refinement through variable removal) applied to (Left) testing points from 2016–2023 (Right) testing points from 2024, Figure S9: Spatial representation of the difference in sample design between our model and similar models. Our model allows for locations with any burn history to be selected as both fire presences and absences for a given year, capturing how interannual variation in conditions influences fire occurrence. Other models effectively only select areas that never burned as fire absences, creating an unintentional sampling bias towards fire resistant areas among the fire absence points, Figure S10: Example layers representing predictor variables from 2024, Table S1: Operational fire probability mapping products produced by government organizations in Thailand, Table S2: Scientific studies presenting approaches to fire probability mapping in peninsular Southeast Asia, Table S3: Predictor variables included in the full model prior to model refinement, Table S4: Error matrix from our independent accuracy assessment of GISTDA burn scar data. Accuracy assessment was done through manual image interpretation in Collect Earth Online, using 90 points generated by randomly selecting 5 fire and 5 non-fire points per year from 2016–2024, Table S5: Recommended avenues for exploring how model performance responds to changes in the years of data the model is developed with and the years of data the model is validated with. For example, the model can be trained and refined on one year of data, or it can be trained and refined on many years of data. Additionally, the model can be validated on the same year(s) of data it was trained and refined with, or it can be validated on different year(s) of data it was trained and refined with. The model is currently trained and refined using fire occurrence and fire predictor data from 2016–2023, and this version can be deployed as is to future years. However, with each new fire season, the model can be retrained and refined with the addition of the most recent fire occurrence and fire predictor data to enhance its performance. Systematically comparing different combinations of developing and validating data can provide insight into how to optimize the model's predictive performance while minimizing the amount of input data required.

**Author Contributions:** Conceptualization, E.B., K.D., K.J., D.M.G.d.T., A.C., K.T., W.S., A.P., V.T., K.W., T.K., M.T., D.G. and D.S.; Data curation, E.B., K.J., A.C., W.S., T.K. and C.S.; Formal analysis, E.B., K.J. and A.C.; Funding acquisition, D.M.G.d.T., K.T., A.P., D.G. and D.S.; Investigation, E.B., K.J., D.M.G.d.T., A.C., W.S., V.T., K.W. and D.G.; Methodology, E.B., K.D., K.J., D.M.G.d.T., A.C., K.T., A.P., V.T., K.W. and T.K.; Project administration, K.D., D.M.G.d.T., K.T., W.S., A.P., C.S. and D.G.; Resources, K.J., D.M.G.d.T., K.T., V.T., K.W., M.T., D.G. and D.S.; Software, E.B., K.J., A.C., T.K. and E.D.; Supervision, K.D., D.M.G.d.T., K.T., A.P., D.G. and D.S.; Validation, E.B., A.C. and E.D.;

Visualization, E.B. and K.D.; Writing—original draft, E.B. and K.D.; Writing—review & editing, E.B., K.D., K.J., D.M.G.d.T., A.C., K.T., W.S., A.P., V.T., K.W., T.K., E.D., C.S., M.T., D.G. and D.S. All authors have read and agreed to the published version of the manuscript.

**Funding:** This research was funded by the World Bank contract number 7208429, solicitation number ECC1284335, "Technical Assistance to Improve Knowledge and Innovative Policy for Wildfire Reduction in Northern Thailand Activity I: Wildfire Risk Map and Management Information System".

**Data Availability Statement:** The original data presented in the study are openly available as a GitHub repository accessible at: https://github.com/enikoebihari/ThailandFireProbability (accessed on 2 September 2025) & https://doi.org/10.5281/zenodo.15935272.

**Acknowledgments:** The authors gratefully acknowledge the support of the World Bank for funding the project titled "Technical Assistance to Improve Knowledge and Innovative Policy for Wildfire Reduction in Northern Thailand", which provided the foundation for this research. We would also like to thank the Thailand Department of National Parks, Wildlife, and Plant Conservation (DNP) and the Regional Community Forestry Training Centre for Asia and the Pacific (RECOFTC) for their close collaboration and data sharing throughout the project. Additional thanks to the Geo-Informatics and Space Technology Development Agency (GISTDA) for providing data, as well as all project stakeholders from local communities, non-profit organizations, and government agencies for their feedback and technical guidance. Note that input from interviews and consultations with individual government officials, field personnel, academic faculty, and community leaders has been deliberately anonymized.

**Conflicts of Interest:** Authors Enikoe Bihari, Karen Dyson, Kayla Johnston, Daniel Marc G. dela Torre, Akkarapon Chaiyana, Karis Tenneson, Wasana Sittirin, Ate Poortinga, Veerachai Tanpipat, Thannarot Kunlamai, Elijah Dalton, Chanarun Saisaward, David Ganz, and David Saah were employed by the company Spatial Informatics Group. The remaining authors declare that the research was conducted in the absence of any commercial or financial relationships that could be construed as a potential conflict of interest.

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
