# Peer review of "Modeling Seasonal Fire Probability in Thailand: A Machine Learning Approach Using Multiyear Remote Sensing Data"

_remotesensing, doi:10.3390/rs17193378_

Round 1

Reviewer 1 Report

Comments and Suggestions for Authors

The paper has strong innovation, solid methodology, and reliable results, but certain methodological details need clarification, and the depth of discussion should be strengthened.

  1. Pages 2-4, lines 62-166: It is recommended to summarize concisely. The introduction is currently not in-depth enough; enhance the comparison of domestic and international research, as well as the necessity and innovation of this study.
  2. Line 169: Do not use the first person "our."
  3. Figure 2: Please label the subfigures as (a), (b), and (c).
  4. Figure 3: The technical roadmap is not standardized; please revise it.
  5. No page numbers throughout the document: Please add them.
  6. Lines 213-257: The sample design needs to clearly explain how spatial autocorrelation is avoided (e.g., buffer distance). Was spatial stratified sampling used, or were negative samples generated artificially?
  7. Lines 329-343: Please specify the dataset partitioning ratio (training/validation/testing) and whether temporal-spatial cross-validation was employed.
  8. Lines 344-392: The results can be expanded upon; clarify the basis for selecting probability thresholds, uncertainty quantification, and comparisons with existing methods.
  9. Lines 393-533: The discussion can be deepened. Why are specific crop types (please specify categories) highly correlated with fire incidents? Is it related to burning agricultural practices? A deeper analysis of the regional cultural and geographical background is recommended.
  10. Lines 393-533: Limitations of the model have not included real-time human activity data (e.g., holiday burning practices). Could this affect short-term predictions? Have the lag effects of climatic variables (e.g., El Niño) been considered?

Author Response

Comment: The paper has strong innovation, solid methodology, and reliable results, but certain methodological details need clarification, and the depth of discussion should be strengthened.

Response: Thank you for your valuable comments, particularly the queries into our methods which have prompted us to develop a more detailed conceptual discussion of certain topics. We carefully considered all of your points and detailed responses are below.

  1. Pages 2-4, lines 62-166: It is recommended to summarize concisely. The introduction is currently not in-depth enough; enhance the comparison of domestic and international research, as well as the necessity and innovation of this study.

Response: Thank you for pointing out where we can strengthen our introduction. We have reorganized our introduction so that we separate our discussions of operational methods and scientific studies, expanding and strengthening both sections with some specific examples. We have added technical information including some additional descriptions of the Thai Fire Danger Rating system, the DNP and GISTDA fire probability maps, (lines 120-133) and the specific scientific publications that include Thailand (lines 139-154). We have also given international examples that use machine learning and temporal data disaggregation in fire probability mapping (lines 190-198). We have more clearly summarized the respective strengths and weaknesses of the different approaches, highlighting the technological gaps in Thailand (lines 169-173, 182-184)

  1. Line 169: Do not use the first person "our."

Response: We have revised the sentence so that it does not use the first person “our.”

  1. Figure 2: Please label the subfigures as (a), (b), and (c).

Response: We have updated this figure to include subfigure labels.

  1. Figure 3: The technical roadmap is not standardized; please revise it.

Response: We have revised the flowchart in accordance with commonly accepted flowchart shape representations for input/output data and processes.

  1. No page numbers throughout the document: Please add them.

Response: Thank you for pointing out this formatting error. We have moved the page numbers back to the right side of the pages.

  1. Lines 213-257: The sample design needs to clearly explain how spatial autocorrelation is avoided (e.g., buffer distance). Was spatial stratified sampling used, or were negative samples generated artificially?

Response: Thank you for the opportunity to clarify our sample design. We have edited the methods to more explicitly describe stratification methods (lines 314-328). Additionally, we have added further discussion to the discussion about spatial autocorrelation, where we recommend and posit the benefits of further stratification based on LULC as future improvements to our current model (lines 667-678).

  1. Lines 329-343: Please specify the dataset partitioning ratio (training/validation/testing) and whether temporal-spatial cross-validation was employed.

Response: Thank you for raising this concern about clarity in our reference data partitioning. We have altered the accuracy assessment section to more explicitly reference data partitioning methods (lines 419-427). We also expanded the data selection section with more specific details about the 80-20% training-validation split and the fact that testing data were generated separately (lines 314-330) (2024 burn scar data were received well after the project had been completed, so these data were not “split” from the initial pool of reference points). We have described more explicitly the temporal stratification in which each year was sampled independently to ensure equal representation of all years in the reference data set. These temporally stratified points were combined into a single data set for the randomized 80-20% training-validation split. The validation data ended up with approximately 116 points for each year, split roughly evenly between fire presence and absence. Thus, reference data were temporally stratified, containing data points representing every year in the study period. The model was trained on data from 2016-2023 and deployed for 2024. There was no spatial stratification beyond the burn scar / no burn scar stratification. 

  1. Lines 344-392: The results can be expanded upon; clarify the basis for selecting probability thresholds, uncertainty quantification, and comparisons with existing methods.

Response: Thank you for the opportunity to make these points clear in our results. We did not threshold our predicted classes for this study, since the desired final products for our work are 1. The model workflow, 2. The associated scripts, and 3. A continuous value probability map of fire risk. These products are used by Thailand government officials for multiple purposes, including resource allocation, and each purpose has different requirements. We added an uncertainty quantification to the methods (lines 424-426), results (lines 487-490), and discussion (lines 507-510, 584-589). We have added text to the methods comparing our approach with other methods (lines 250-272). Additionally, the discussion now provides in-depth comparisons between our model and other methods, and the reasons for differences in these patterns (512-622).

  1. Lines 393-533: The discussion can be deepened. Why are specific crop types (please specify categories) highly correlated with fire incidents? Is it related to burning agricultural practices? A deeper analysis of the regional cultural and geographical background is recommended.

Response: Thank you for providing the opportunity to emphasize this in the discussion. We have added a paragraph in the discussion (lines 526-545) to link this back to its initial discussion in the introduction (lines 72-103). 

  1. Lines 393-533: Limitations of the model have not included real-time human activity data (e.g., holiday burning practices). Could this affect short-term predictions? Have the lag effects of climatic variables (e.g., El Niño) been considered?

Response: Thank you for pointing out these directions for future improvement of the model. We have incorporated them into the discussion (lines 566-569, 683-685).

Reviewer 2 Report

Comments and Suggestions for Authors

This manuscript aims to present seasonal fire probability mapping methodology using a Random Forest (RF) machine learning model in the Google Earth Engine (GEE) platform. The manuscript trained the model on historical fire occurrence and fire predictor layers from 2016-2023, and applied it to 2024 conditions to generate a probabilistic fire prediction. The study is interesting but lacks innovation.

  1. The introductory section currently lacks logical coherence and should instead provide a systematic synthesis of existing research relevant to the study's core theme. A critical classification and evaluation of prevailing methodologies is required, with explicit analysis of their respective strengths and limitations, ultimately justifying the necessity of the proposed research.
  2. Figure 1 lacks clear conceptual significance, with the right panel merely displaying a remote sensing image that fails to establish logical coherence with the left-side textual content. Diagrammatic representations should maintain conciseness, and non-essential annotations should be omitted.
  3. In line 181, the standardized abbreviation "NDVI" in remote sensing must be expanded to its full technical designation: ‌N‌ormalized ‌D‌ifference ‌V‌egetation ‌I‌ndex.
  4. The model selection exclusively employs the Random Forest algorithm without detailed justification; a single sentence on line 193 merely states its effectiveness. This precludes ascertaining the scientific justification for the choice, raising questions about whether alternative models yielded inferior results. The selection appears arbitrarily based solely on the algorithm's performance with remote sensing imagery.
  5. The Random Forest model represents a conventional machine learning approach, yet the manuscript contains no description of any innovative modifications or enhancements to this algorithm. It is merely applied for conventional implementation and result presentation, indicating a lack of substantial innovation in this research.
  6. Figure 3 employs an oversimplified representation, presenting three identical tree diagrams despite utilizing the Random Forest model. This redundant visualization fails to elucidate the model's fundamental principles, particularly given the ensemble nature of the methodology.
  7. The results analysis remains superficial, necessitating comparative validation across multiple modeling algorithms. Quantitative metrics should be employed to demonstrate the superiority of the selected model in this manuscript.
  8. The Conclusions section should not merely restate the results; it requires restructuring its logical framework for an enhanced presentation.

Author Response

This manuscript aims to present seasonal fire probability mapping methodology using a Random Forest (RF) machine learning model in the Google Earth Engine (GEE) platform. The manuscript trained the model on historical fire occurrence and fire predictor layers from 2016-2023, and applied it to 2024 conditions to generate a probabilistic fire prediction. The study is interesting but lacks innovation.

Response: Thank you for your detailed comments. Your feedback has encouraged us to restructure and supplement our text to enhance its logical coherence and deepen its arguments. We respectfully note that while our study does employ a well-established machine learning algorithm, the innovation lies in how we adapt and operationalize random forest for fire probability mapping in Thailand. We would like to emphasize some key improvements: empirical variable selection, disaggregation of historical data to capture interannual variability, and representative sample design agnostic to burn history. Additionally, our framework was codeveloped with local fire management agencies to ensure operational usability. These elements distinguish our work from existing studies in the region and represent important methodological advances beyond a conventional application. We carefully considered all of your points and detailed responses are below.

  1. The introductory section currently lacks logical coherence and should instead provide a systematic synthesis of existing research relevant to the study's core theme. A critical classification and evaluation of prevailing methodologies is required, with explicit analysis of their respective strengths and limitations, ultimately justifying the necessity of the proposed research.

Response: Thank you for the opportunity to expand and reorganize our introduction to make it more logically coherent and provide a stronger argument for the need of our research. We have reorganized the introduction so that we separate our discussions of operational methods and scientific studies, expanding and strengthening both sections with some specific examples. We have added technical information including some additional information on the Thai Fire Danger Rating system, the DNP and GISTDA fire probability maps, and the specific scientific publications that include Thailand (lines 120-133, 139-154). We have clearly summarized the respective strengths and weaknesses of the different approaches, and discuss the tradeoffs between model types. We have given some additional international examples that use machine learning and temporal data disaggregation in fire probability mapping (lines 190-198). We have also more clearly referenced Supplement Tables S1 and S2.  These tables provide the technical details of the studies from our systematic literature review on the topic of fire probability mapping in Southeast Asia. Building on this, we more clearly discuss the potential improvements to be made in both sample design and data preprocessing, and directly state how our study addresses the limitations of previous work to justify the necessity of our work (lines 157-160,169-173,182-184, 199).

  1. Figure 1 lacks clear conceptual significance, with the right panel merely displaying a remote sensing image that fails to establish logical coherence with the left-side textual content. Diagrammatic representations should maintain conciseness, and non-essential annotations should be omitted.

Response: Thank you for pointing out the design flaws and low relevance of this figure to the key themes of the manuscript. We have removed it from the manuscript.

  1. In line 181, the standardized abbreviation "NDVI" in remote sensing must be expanded to its full technical designation: ‌N‌ormalized ‌D‌ifference ‌V‌egetation ‌I‌ndex.

Response: Thank you for pointing out this error. We have added the full technical term.

  1. The model selection exclusively employs the Random Forest algorithm without detailed justification; a single sentence on line 193 merely states its effectiveness. This precludes ascertaining the scientific justification for the choice, raising questions about whether alternative models yielded inferior results. The selection appears arbitrarily based solely on the algorithm's performance with remote sensing imagery.

Response: Thank you for raising this concern about rationale for model selection. We have restructured and added text to the methods to more explicitly state our reasons for why random forest was selected over other methods (lines 250-272) (namely: the lower need for hyperparameter tuning than more complex models, the availability of a built-in variable importance metric, and the robustness to noisy multidimensional data compared to simpler models).  For this work, we needed a framework that was both reliable and operationalizable, and model performance was not the only criterion for our decision. Usability and interpretability were equally important, especially considering the operational context of our work. The model selection decisions we made were heavily guided by the input of local scientists, industry professionals, and government officers during our interviews and codevelopment workshops in Thailand. We aimed to create a workflow that would be easily learned and utilized by fire experts in Thailand without continual external support. Resources and knowledge surrounding machine learning and cloud computing are extremely limited within the fire management agencies in Thailand, so we needed a workflow based on a well-known and reliable performer that would need little long term maintenance from external consultants. The remote sensing community recognizes random forest as a consistently high performing classification model; this reliability is particularly critical for our goals of scaling and refining the study in both the spatial and temporal dimensions. 

While we understand the reviewer’s concern that our workflow did not directly test and select the model with the highest AUC in this specific pilot study area, this was not the goal of this specific work. However, it would be a valuable addition for the future, and we have added this as a suggestion to the discussion (lines 619-622) where we discuss future improvements. 

  1. The Random Forest model represents a conventional machine learning approach, yet the manuscript contains no description of any innovative modifications or enhancements to this algorithm. It is merely applied for conventional implementation and result presentation, indicating a lack of substantial innovation in this research.

Response: Thank you for the opportunity to clarify how we have contributed to the literature and improved fire probability mapping in Thailand. We believe that our work is a significant contribution to the literature on fire probability mapping in peninsular Southeast Asia, both in our use of machine learning and in our specific improvements on sample design and model refinement. 

The development of a machine learning-based model is itself a step forward in Thailand, as all operational methods and most scientific studies rely on multi-criteria analysis (MCA). Even in scientific literature, we could only find one Thailand-focused study which applied machine learning, and this used only Bayesian Networks (BN) and Naïve Bayes (NB) (Phoompanich et al. 2019). The only other machine-learning based work which included Thailand was a study that spanned Southeast Asia (He et al. 2021). We have added text that more explicitly states this (lines 139-154, 169-173). 

Additionally, unlike other studies, we empirically refine predictor variables from a comprehensive set of initial variables. Further, to our knowledge, our methodology is the only one that samples each year separately when generating reference points and predictor layers, allowing us to account for interannual variation in the data. This is in stark contrast to other studies which aggregate all fires and predictors into single layers, producing a single map that does not reflect the variation of conditions between fire seasons. This annual sampling also allows us to reduce bias in our reference points, as areas can be selected as fire absences or presences for each year regardless of whether they burned in the past or future. This is not possible in the other studies which effectively only sample fire absences from areas that never burned in the study period, entirely missing the nuance that fire absence also occurs in areas that may burn in other years. (lines 216-224, 512-622)

  1. Figure 3 employs an oversimplified representation, presenting three identical tree diagrams despite utilizing the Random Forest model. This redundant visualization fails to elucidate the model's fundamental principles, particularly given the ensemble nature of the methodology.

Response: Thank you for pointing out that this representation oversimplifies random forest models. We have replaced the image with a new diagram that explicitly depicts unique decision trees and voting.

  1. The results analysis remains superficial, necessitating comparative validation across multiple modeling algorithms. Quantitative metrics should be employed to demonstrate the superiority of the selected model in this manuscript.

Response: Thank you for highlighting the need for a more explicit quantitative comparison of models in the main text of the manuscript. We have added a comparison in AUC between our model and similar ML models, as well as justification for why we have not quantitatively compared ML and MCA accuracies (lines 476-486). A more detailed breakdown of the accuracy assessment methods for the cited studies in Southeast Asia can be found in Supplement Table S2. In the discussion, we further discuss why our AUC is slightly lower than some other ML models (lines 590-622). This is likely the result of our methods better accounting for temporal variation and drawing a more representative sample, capturing more variability in the training and validation data and minimizing bias that may have artificially inflated the AUCs of other studies (lines 615-616). 

Additionally, we discuss that random forest was chosen not only for its reliable performance but also its simplicity to understand and implement in an operational setting (lines 250-272). We chose a model that would be accessible within the cultural, political, and technical landscape of Thailand’s fire management agencies, purposefully avoiding more complicated models. Random forest is easy to conceptualize and performs well even with minimal parameter tuning, and is easy to implement in many spatial analysis platforms. This is the first random forest fire probability model trained and refined specifically for Thailand with local data, and it provides significant technical advances for the region, where MCA methods are currently the standard practice for the government.

  1. The Conclusions section should not merely restate the results; it requires restructuring its logical framework for an enhanced presentation.

Response: We have restructured and revised our conclusion to better present our points and their implications. We have added text summarizing how our methodology can be scaled and how it can improve fire management in Thailand (lines 691-727).

Reviewer 3 Report

Comments and Suggestions for Authors

1.A brief summary

The article presents a novel seasonal fire probability mapping methodology for Thailand using Multi-source free data and Random Forest(RF) machine learning model in the Google Earth Engine(GEE) platform. The predictor variables cover a wide range of factors, including topography, fuels, potential fire behavior, forest type, vegetation characteristics, climate, water availability, crop type, recent burn history, and human influence and accessibility, which has substantial scientific significance for understanding and predicting seasonal fires in Thailand. The method represents a significant advancement over traditional approaches that rely heavily on subjective multi-criteria analysis(MCA) and static data aggregation methods, which holds significant practical value in assessing the risk of forest fires and in taking reasonable measures to mitigate such risks.

2.General concept comments

(1)The original resolution of the predictor variables used in this study ranged from tens of meters to several kilometers. Finally, the authors resampled them all to 300 meters. The authors need to explain the reason for this.

(2)The training and validation reference data(Fire Presence and Absence Data)used by the author is derived from burn scar data for 2014-2023 from GISTDA. This data itself has certain errors (with an accuracy of 66.8%), which brings considerable uncertainty to the research results. It is recommended that the author verify these data effectively before using them as reference data (such as data verified by multiple fire data products), in order to increase the credibility of the research results.

(3)In the introduction, the author mentioned commonly used machine learning methods including Random Forest (RF), Bayes Network (BN), Naive Bayes (NB), and Support Vector Machine (SVM). Although the author justified the rationality of the RF method in the methods section, they did not further explain whether the other methods would be better compared to the RF method. This point needs to be further clarified by the author.

(4)The author uses the RF classification method to predict the presence or absence of fire. According to the RF classification results, a categorical variable (Fire Presence or Absence) should be generated. The author needs to further explain how the fire probability is derived. Is it based on the regression prediction results of RF? This point needs to be further clarified in the method section.

3.Specific comments

(1)In Figure 2, the study area appears to be in the northwest of Thailand. However, the author wrote "the 9 northeasternmost provinces of Thailand". Should it be the nine provinces in the far northwest?

(2)The "" in line 44 is redundant and should be removed.

(3)In Table 1, the NDMI and EVI values in the "Vegetation Characteristics" section do not exist in the dataset provided in Reference 111. The author needs to provide the calculation formulas.

Author Response

1.A brief summary

The article presents a novel seasonal fire probability mapping methodology for Thailand using Multi-source free data and Random Forest(RF) machine learning model in the Google Earth Engine(GEE) platform. The predictor variables cover a wide range of factors, including topography, fuels, potential fire behavior, forest type, vegetation characteristics, climate, water availability, crop type, recent burn history, and human influence and accessibility, wsurrohich has substantial scientific significance for understanding and predicting seasonal fires in Thailand. The method represents a significant advancement over traditional approaches that rely heavily on subjective multi-criteria analysis(MCA) and static data aggregation methods, which holds significant practical value in assessing the risk of forest fires and in taking reasonable measures to mitigate such risks.

Response: Thank you for your encouraging and insightful comments. We appreciate your detailed look at our manuscript and identification of several important opportunities to improve it - particularly regarding the spatial and statistical implications of some of our methodological decisions. We carefully considered all of your points and in depth responses are below.

2.General concept comments

(1)The original resolution of the predictor variables used in this study ranged from tens of meters to several kilometers. Finally, the authors resampled them all to 300 meters. The authors need to explain the reason for this.

Response: Thank you for highlighting the need to clarify this point. We have added text to address this (lines 356-359). We resampled all predictor variables to a common 300m resolution to ensure spatial alignment across layers of different native resolutions. We selected 300m as a compromise between the wide range of resolutions. We sampled all terrestrial layers up to the 300m of the lowest resolution terrestrial layers (the Pettinari & Chuvieco fuel layers), as is best practice in remote sensing, but sampling up to the 4.6km resolution of the climate data would have eliminated most fine-scale ecological variation in these layers. Thus, we chose to downsample and smooth the climate layers to reduce blocky artifacts of the original data set. We also recommend that all global layers be replaced with local high-resolution data sets from the Thailand Meteorological Division during operational use of these methods (lines 514-520, 653)), which would eliminate the need for reconciling 4.6km resolution data with sub-1km resolution data .

(2)The training and validation reference data(Fire Presence and Absence Data)used by the author is derived from burn scar data for 2014-2023 from GISTDA. This data itself has certain errors (with an accuracy of 66.8%), which brings considerable uncertainty to the research results. It is recommended that the author verify these data effectively before using them as reference data (such as data verified by multiple fire data products), in order to increase the credibility of the research results.

Response: Thank you for raising this important concern. We have conducted a small independent accuracy assessment of the burn scar data (as resources, time, and manpower allowed), and have added these results to the mascript (lines 294-295) and as Supplement Table S4. We would like to highlight that the GISTDA burn scar data were the highest quality burn scar data our research team was able to acquire (lines 287-290), and we did not have the resources to generate a full reference dataset ourselves. Other agencies in Thailand do produce manually delineated burn scar polygons, which likely have extremely high accuracy, but these data are highly confidential and cannot be shared outside of the relevant agencies. We ran into data accessibility issues with many of the necessary layers, so we focused on creating a proof-of-concept type workflow that allows for easy substitution of data with higher quality local data layers. This will allow practitioners to create higher quality end products for operational use. We also recommend substituting manually delineated burn scar data for the training and validation reference data (lines 644-652, 663-666).

(3)In the introduction, the author mentioned commonly used machine learning methods including Random Forest (RF), Bayes Network (BN), Naive Bayes (NB), and Support Vector Machine (SVM). Although the author justified the rationality of the RF method in the methods section, they did not further explain whether the other methods would be better compared to the RF method. This point needs to be further clarified by the author.

Response: Thank you for the opportunity to clarify our rationale for model selection. We have restructured and added text to our methods (lines 250-272) to more explicitly state our reasons for why random forest was selected over other methods (namely the lower need for hyperparameter tuning than more complex models, the availability of a built-in variable importance metric, and the robustness to noisy multidimensional data compared to simpler models). We summarize these advantages rather than individually comparing RF to each model. We needed a framework that was both reliable and operationalizable, and model performance was only one criterium for our decision. Usability and interpretability were equally important, especially considering the operational context of our work. The decisions we made were heavily guided by the input of local fire scientists and managers during our interviews and codevelopment workshops in Thailand. We aimed to create a workflow that would be easily learned and utilized by fire professionals in Thailand without continual external support. Resources and knowledge surrounding machine learning and cloud computing are extremely limited within the fire management agencies in Thailand, so we needed a workflow based on a well-known and reliable performer that would need little long term maintenance from external consultants. The remote sensing community recognizes random forest as a consistently high performing classification model; this reliability is particularly critical for our goals of scaling and refining the study in both the spatial and temporal dimensions.

(4)The author uses the RF classification method to predict the presence or absence of fire. According to the RF classification results, a categorical variable (Fire Presence or Absence) should be generated. The author needs to further explain how the fire probability is derived. Is it based on the regression prediction results of RF? This point needs to be further clarified in the method section.

Response: Thank you for pointing out the need to clarify this point. We have rephrased the methods to more explicitly state how probability was calculated (lines 278-279). The probability at each pixel was the percentage of the 500 decision trees that voted for fire presence at that pixel. Google Earth Engine’s smileRandomForest().setOutputMode() function allows users to automatically calculate RF outputs as probabilities this way.

3.Specific comments

(1)In Figure 2, the study area appears to be in the northwest of Thailand. However, the author wrote "the 9 northeasternmost provinces of Thailand". Should it be the nine provinces in the far northwest?

Response: Thank you for pointing out this error, we have corrected it in all four locations in the text.

(2)The "" in line 44 is redundant and should be removed.

Response: We are unable to find this error in the text at line 44. Please let us know if we are missing it.

(3)In Table 1, the NDMI and EVI values in the "Vegetation Characteristics" section do not exist in the dataset provided in Reference 111. The author needs to provide the calculation formulas.

Response: Thank you for pointing out the need for providing the formulas in the main text. We have added the NDMI and EVI formulas into Table 1. A more detailed breakdown of all formulas used in data preprocessing steps in Supplement Table S3.

Round 2

Reviewer 2 Report

Comments and Suggestions for Authors

The authors have addressed the queries raised, acknowledging the limitations of existing research in Thailand as noted in their response. While this manuscript contributes to local applications in Thailand, the study merits further in-depth exploration. I have no additional concerns. References 1 and 37 require verification due to inconsistent language presentation (Thai and English), and references 69, 165, and 166 should be checked for compliance with formatting conventions.

Author Response

Thank you for pointing out these citations. We have updated all of them according to MDPI's citation formatting standards with the full information available about them.